# Application of a Generic Participatory Decision Support System for Irrigation Management for the Case of a Wine Grapevine at Epirus, Northwest Greece

Ioannis L. Tsirogiannis [1,*], Nikolaos Malamos [2] and Penelope Baltzoi [1]

1    Department of Agriculture, University of Ioannina, Kostakii Campus, 47100 Arta, Greece
2    Department of Agriculture, University of Patras, Nea Ktiria Campus, 30200 Messolonghi, Greece
*    Correspondence: itsirog@uoi.gr

**Abstract:** In southern Europe, irrigation is the major water user and thus, development of operational tools that support decisions aiming to improve irrigation management, is of great importance. In this study, a web-based participatory decision support system for irrigation management (DSS), based on the principles of UN FAO's paper 56, without requirement for any special monitoring hardware to be installed in each field, is evaluated for the case of a commercial wine grapevine (*Vitis vinifera* 'Vertzami') located at Epirus (northwest Greece), for two successive years (2021 and 2022). The soil moisture time series that were generated by the DSS's model were compared to those measured by soil moisture sensors. The Mean Absolute Error (MAE) and Root Mean Square Error (RMSE) ranged between 2.98–3.22% and 3.63–4.06%, respectively, under various irrigation practices and goals. Irrigation resulted very high yields and Crop Water Productivity (WPC) was 20–44% improved when following the DSS's recommendations. The results also confirm potential pitfalls of sensor-based soil moisture monitoring and rainfall estimations using mathematical models. Finally, the value of water meters as practical sensors, which could support efficient irrigation management, is underlined. In every case, mindful application of decision support systems that require minimum or no hardware to be installed in each field, could extensively support growers and agronomic consultants to test, document and disseminate good practices and calculate environmental indices.

**Keywords:** vertzami wine; clay loamy soil; water requirements; efficient irrigation; micro-irrigation; soil moisture sensors; dielectric capacitance sensors; DSS; water productivity

## 1. Introduction

Efficient soil water management is an important goal in viticulture under Mediterranean type climatic conditions, where seasonal drought is a common phenomenon and soil and atmospheric water deficits, high evapotranspiration demand and water availability can exert significant constraints regarding the quantity and quality of yield. This goal is significantly enhanced by climatic changes that promote water scarcity and irrigation could play a crucial role to mitigate relevant challenges [1–4]. In southern Europe, where traditionally grapevine is a non-irrigated crop, a continuous increase of the area of irrigated vineyards is reported during the last decades [5–9]. In this framework, several decision support systems that provide functions for irrigation management in vineyards have been applied during the last decade [10–12]. Some of those systems include grapevines in the sets of crops for which they have been evaluated and some were developed with a focus on that specific crop. The provided solutions ranged from inclusion of sensor based soil moisture monitoring in vineyard management platforms [13], systems that were oriented to irrigation management and werPlease check that All References are mentioned in a numerical order in the main text and revise if needed.

Combined with weather and soil moisture sensors placed at each vineyard [14], systems that provided integrated vineyard management options for large areas using limited

number of weather stations [15] and systems that could operate efficiently using only data from weather models and satellite remote sensing imagery [16]. While the basis for the irrigation scheduling calculations was in most cases the relevant UN's Food Agriculture Organisation (FAO) guidelines, a wide range of modelling approaches has been reported, from deterministic operations [14–16] to application of artificial intelligence [17]. Interesting trade-offs regarding the perspectives of adoption of such tools by growers and agronomic consultants in commercial cropping systems are linked to the relationships between data entry needs, model complexity, requirements for manual data inputs and automatic monitoring [10]. In every case, sustainable vineyard water management requires reliable, easy-to-use, and cost-effective tools, that can adapt to the specific environmental conditions and management strategies of each case and provide "real-time" recommendations regarding 'when' to irrigate and 'how much' water to supply.

The present study concerns the evaluation for the case of 'Vertzami' wine grape of a generic web-based participatory decision support system for irrigation management, which is based on UN's Food Agriculture Organisation (FAO) guidelines for the determination of ETo and soil water depletion and does not require any specialized hardware to be installed in each field. The aim of the evaluation was to reveal and discuss advantages and probable pitfalls of decision support systems for irrigation management that do not require installation of sensors at each field, to conclude whether the DSS under evaluation could be used as an alternative to soil moisture sensors, and if its application has the potential to improve water productivity for grapevine.

## 2. Materials and Methods

The evaluation was carried out during two consecutive growing seasons (March–September 2021 and 2022) in a commercial vineyard at the plain of Arta, Epirus, northwest Greece (39.18584° N, 20.97479° E (WGS84), altitude 45 m), planted with 6 years old, *Vitis vinifera* var. 'Vertzami', a red wine grape variety. The total area of 'Vertzami' in Greece is 175.60 ha, most of which are found at the Ionian Islands, Western Greece and Epirus [18]. 'Vertzami' cultivation is typically rainfed. It becomes fully ripped at mid–September and yields up to 6–7 t ha$^{-1}$, but it can reach 15 t ha$^{-1}$ [19].

The climate of the area is of Mediterranean type with moderate winters and hot summers. The long-term average annual temperature is 17.2 °C and the long term mean annual precipitation is 1084 mm (from which 230 mm concern the period from April up to September). Although winters are abundantly rainy, the area experiences, especially during the last years, prolonged drought periods at summer and the first half of autumn, rendering irrigation necessary to meet crops' water needs [20].

The soil texture of the vineyard, down to 60 cm was of clay loamy (CL) type (31.4% sand, 31.4% silt and 37.2% clay), and the soil pH was 6.4. The irrigation parameters of the soil were determined though its water retention curve (WRC), by applying the Haines funnel approach, as follows (volumetric %): saturation ($\Theta$s) = 45% [21], field capacity (FC) 34% [22] and wilting point (WP) = 13% [21].

Water was supplied by the Land Reclamation and Irrigation Water Management Organisation (LRO) Grammenitsa-Vlaherna. Its pH was 8.1 and its EC was 0.44 dS m$^{-1}$. The irrigation period for this LRO spaned from the beginning of June up to the middle of September, while water for the irrigation sector of the vineyard under consideration was usually available every second or third day.

According to the Greek legislation [23], the generic limits for irrigation water usage for grapes in Epirus at northwest Greece, range between 4370 and 5340 m$^3$ ha$^{-1}$ (referred to the total area of the field) for an irrigation period spanning from 15 April to 15 September, for zero contribution by rain and application of water via a micro-irrigation system with irrigation efficiency = 90%, for the case of a LRO that distributed water using a well maintained closed-pipes distribution system.

Irrigation at the vineyard was performed using one adjustable (0–70 Lh$^{-1}$) flow dripper per vine. An irrigation system audit showed that the average flow per dripper was 12.5 Lh$^{-1}$ (standard error 0.55 Lh$^{-1}$). The uniformity of the system (Us) was found equal to 75.4% [24].

The total area of the experimental site was 0.04 ha, and it hosted 14 rows of 15–16 vines each (214 plants in total). Vines were trained on a vertical trellis system of 1.20 m height. All cropping practices (pruning, fertilisation, plant protection etc.) were identical for the whole experimental site, except irrigation scheduling. During 2021 the whole experimental site, consisted of one plot that was irrigated according to grower's experience, which reflected typical local practices (indicated as GRO practice or plot hereafter), while during 2022, the experimental area was divided in two plots of equal size, one of which continued practice GRO, while the other (indicated as DSI practice or plot hereafter) was irrigated following recommendations generated by IRMA_SYS (IRMASYS P.C., Igoumenitsa, EP, Greece), an operational generic web-based decision support system for irrigation management (the DSS hereafter) that covers many LROs in Greece. The DSS does not require the installation of any special sensor at each field and provides real-time forecasts for soil moisture at the end of each day and generates recommendations for future irrigation applications, based on the outcomes of a daily ETo (Penman-Monteith) and water balance model that follows the principles of FAO's paper 56 [25,26]. The DSS considers: (a) measurements of weather parameters from reference automatic agro-meteorological stations for each area; (b) soil, crop and irrigation system parameters; (c) time and volume of the actual irrigation applications and (d) weather data forecasting. The DSS caters generic sets of parameters for each irrigation system type, soil [27] and crop [26] and suggests their adjustment for specific conditions. Irrigation is recommended by the system when soil moisture is estimated to have reached the lower level of the readily available water (RAW), while the irrigation dose is controlled by a refill factor (RF). Documentation and analytical flowcharts of the algorithm that is followed by the DSS, is provided by Malamos et al. [28]. When RF is set to 1, the goal of each irrigation recommendation is to refill soil moisture up to the field capacity. In case that no salinity problems exist, it is generally prudent to use RF values of a bit less than 1 to avoid probable excess of FC. The recommendations of the DSS are applied manually as it does not support automatic control of water valves.

The DSS that covers the LRO's of the plain of Arta (about 20,000 ha, https://arta.irmasys.com/, accessed on 15 November 2022) uses weather timeseries from seven automatic agro-meteorological stations of the Open Hydrosystem Information Network (OpenHi.net, https://system.openhi.net/, accessed on 15 November 2022) [29].

For monitoring soil moisture, (3) three dielectric capacitance sensors (type 10 HS, METER Group Inc., Pullman, WA, USA) were placed for each plot, resulting a density of one sensor per 128 and 64 m$^2$ of total field area, for 2021 and 2022 respectively. The sensors were placed 0.10 m away from drippers, at the middle of the length of lateral pipes and at a depth of 0.20 m, following relevant recommendations for electromagnetic sensors [30] The generic equation provided by the manufacturer for calculating volumetric water content in mineral soils was used (accuracy ±0.03 m$^3$ m$^{-3}$). The time series were processed for outliers to be removed [31]. Soil moisture was considered uniform through the whole soil depth under consideration. For both years, soil moisture measurements during the irrigation period were compared to estimations of soil moisture generated by the DSS.

Water usage by the irrigation system was measured using one 25 mm volumetric dry dial water meter (accuracy 1 L, type DS-TRP, Maddalena S.P.A., Povoletto, UD, Italy) per plot. A rain sensor (ECRN 100 rain gauge, Meter Group, Inc., Pullman, WA, USA), plugged on an automatic datalogger (em50, Meter Group, Inc., Pullman, WA, USA) was also installed at the experimental site to monitor rain and evaluate the relevant estimations of the DSS.

The water status of grapevines was monitored during 2022, by measurements of midday Leaf Relative Water Content (LRWC) using 3 top-most fully expanded healthy leaves from 4 random vines per plot, collected at solar noon. The samples were transported in sealed insulated boxes, weighed to obtain the fresh weight (FW) and then 8 leaf disks

(total area 10 cm$^2$), were soaked in water for 4 h and weighed to obtain the turgid weight (TW). The turgid leaves were oven-dried at 75 °C for 24 h in a ventilated oven, and then weighed (DW). LRWC (%) resulted from the ratio [(FW-DW)/(TW-DW)] × 100 [32,33].

The grapes were harvested following commercial harvest conditions. For 40% of the vines, randomly selected, the grapes were weighted to estimate total yield per plot.

The ratio between grape yield (GY) and total water use (TWU) amount from both irrigation and precipitation from bud break to harvest in each year, was used to quantify Crop Water Productivity (WPC, kg ha$^{-1}$/m$^3$ ha$^{-1}$) [34].

To evaluate the performance of the DSS to model soil moisture at the end of the day, Mean Absolute Error (MAE) and Root Mean Square Error (RMSE) were used [35,36]. MAE corresponds to the average absolute difference between the DSS model prediction and the measurements of volumetric soil moisture at the end of each day. RMSE is a quadratic scoring rule that also measures the average magnitude of the error. It's the square root of the average of squared differences between prediction and actual observation. MS-Excel (Microsoft Corp, USA) was used for the processing and the statistical analysis of the data.

## 3. Results and Discussion

### 3.1. Agrometeorological Parameters

Figure 1 presents the reference evapotranspiration (ETo) and the rain for 2021 and 2022. The ETo values are those that were calculated from the DSS, while for rain, both the values from the DSS and those from the rain gauge that was installed at the vineyard are presented. The reason is that previous reports showed that, while the estimation of ETo by the DSS for the area where the vineyard was sited is adequate, rain was not sufficiently estimated. The results showed that from 15 March up to 8 September 2021, the DSS estimated 111 mm of rain while the sensor at the vineyard counted 240 mm. Additionally, from 15 March up to 24 September 2022, the DSS estimated 60 mm of rain while the sensor at the vineyard counted 156 mm. While estimates of daily reference evapotranspiration (ETo) in agricultural areas, via spatial interpolation of agrometeorological parameters that are monitored by a network of stations, generally provide good results [37,38], deviations between predicted and actual daily rainfall values is a common pitfall for such approaches [39]. As rainfall is a basic parameter of irrigation water balance, verification is recommended for rainfall data that are based on measurements which are made away from the field under consideration. For the present evaluation, the DSS model was run using the precipitation data from the rain gauge that was installed at the vineyard.

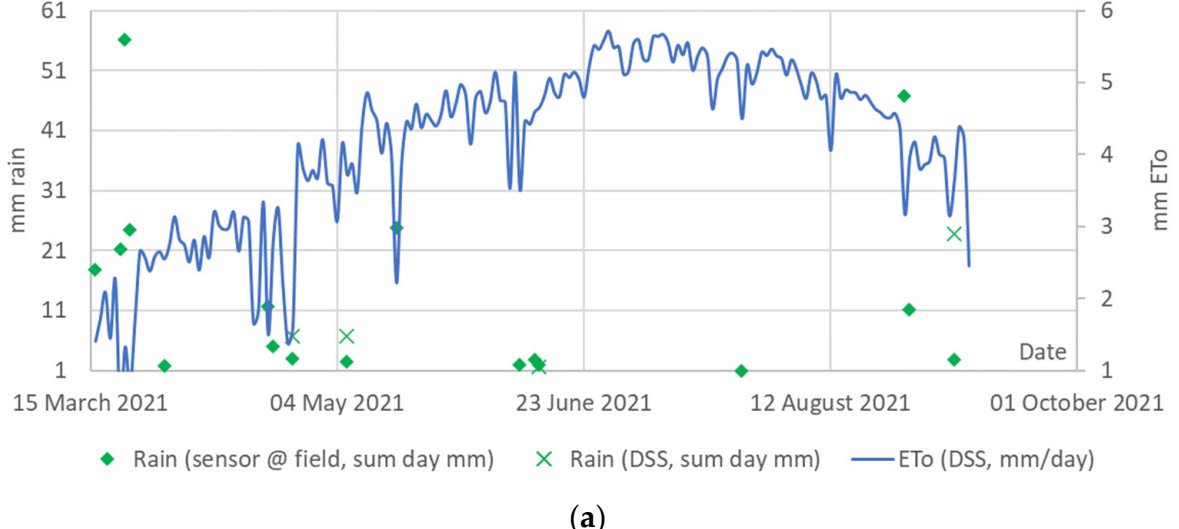

(a)

**Figure 1.** *Cont.*

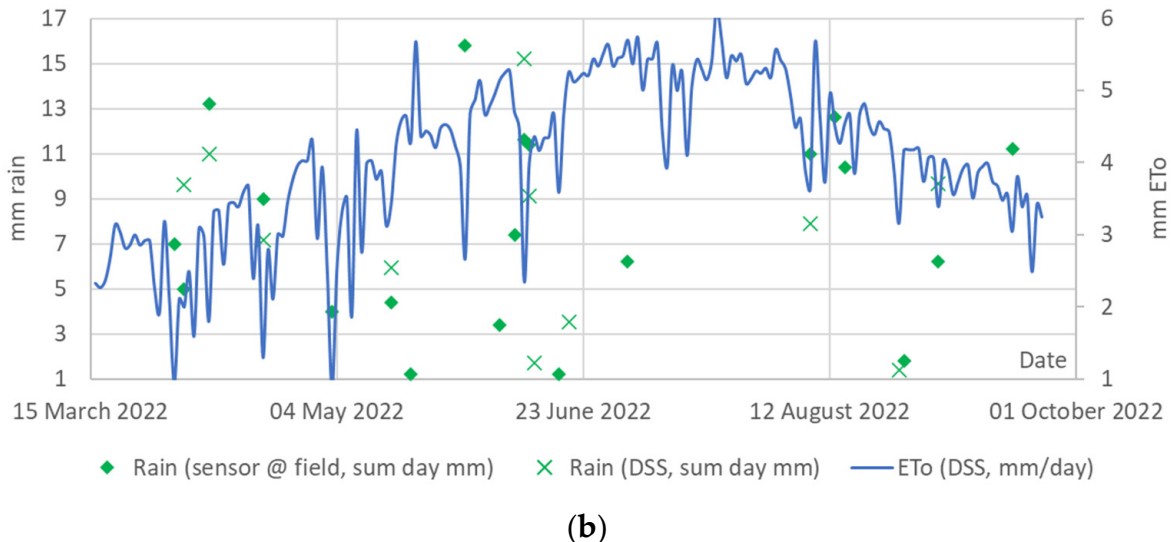

**(b)**

**Figure 1.** ETo according to DSS and rain according to DSS and actual sensor installed at the field during 2021 (**a**) and 2022 (**b**).

*3.2. Measured Soil Moisture*

To ensure the operational performance of a DSS for a specific crop and cropping conditions, evaluation under the commercial field conditions is required [10]. Several studies that investigated the use of electromagnetic sensors in novel automated irrigation management applications presented very good results, under the condition that optimal sensor positioning has been achieved [30,40–43]. In this framework, the use of averaged values from a reasonable number of soil moisture sensors that are characterised by good manufacturing quality and well proved efficiency, installed by experienced professionals and are continuously supervised, is generally expected to provide adequate estimations of soil moisture levels [30,40].

The proposal of a suitable position for soil moisture sensors for micro-irrigation systems is a complex task, while several drawbacks and pitfalls may arise when they are used in commercial open fields, due to the heterogeneity of soil properties, the formation of a root system pattern, and the evolution of a soil moisture profile during and after irrigation and rain events [30,40].

At the present evaluation, a dense network of sensors was deployed, as one soil moisture sensor was installed every 128 m$^2$ of total field area for 2021 and 64 m$^2$ for 2022, respectively. Nevertheless, significant ranges of soil moisture values were monitored as can be seen in Table 1, which presents data regarding average differences at the measured values of the soil moisture sensors. It must be noted that, even in such cases, the differences could be quite significant and in-situ measurements of soil moisture could easily become problematic. Additionally, during the whole evaluation period, 50% of the sensors that were used had to be repaired or substituted, because of functionality problems and damages by animals and people who worked at the vineyard.

In every case, because of the straightforward understanding of the measurements of electromagnetic soil moisture sensors by growers and agronomic consultants [41,44], it is reasonable to evaluate the performance of a decision support system for irrigation management by its capability to estimate measurements that are made by such hardware [45].

**Table 1.** Measured soil moisture (volumetric average, minimum and maximum values) at the end of each day, plus expected accuracy limits according to the manufacturer (the values in the parenthesis are the standard errors).

| Year/Plot | Average Difference between Maximum and Minimum Measured Values from Soil Moisture Sensors (%) | Average Difference between Maximum Plus the Accuracy Limit of 3% and Minimum Minus the Accuracy Limit of 3% Measured Values from Soil Moisture Sensors (%) |
|---|---|---|
| 2021 GRO | 7.39% (0.37) | 13.38% (0.37) |
| 2022 DSI | 7.90% (0.12) | 13.90% (0.12) |
| 2022 GRO | 3.74% (0.19) | 9.74% (0.19) |

*3.3. Adjustment and Evaluation of the DSS*

The values for parameters of the DSS were based on generic literature values for wine grapes [26,46–48] and information that was estimated by the grower or an agronomic consultant. The DSS's parameters for each year (2021 and 2022) are presented in Table 2. The DSS model considers as potential effective rain (rain that reaches the soil surface), 80% of the measured rainfall. The limits for the wetted area and the irrigation efficiency were estimated by performing an irrigation audit. The maximum allowable depletion of the available soil moisture (MAD, % of total available water (TAW = FC − WP)) which is a managerial parameter that sets the lower level of the readily available water (RAW) was set to 45% for 2021, the generic value proposed by the system for wine grape [26], and 52% for 2022 in order to exert a bit of extra water stress to the plant in respect to the grower's practice for impacting grape's sugar content. The limits of root depth were estimated based on the grower's observations. The planting date estimation was based on the budburst date of each year. The Kc stages duration and values were based on the generic values proposed for wine grape adapted to conditions in Greece [26,46–48].

Regarding agrometeorological conditions, ETo was used as estimated by the DSS, while the precipitation measurements from the actual sensor (rain gauge) at the vineyard were used to run the DSS model. The data (date and volume of water as measured by the water meter) of the actual irrigation applications that have been performed were manually registered to the DSS. As it was expected [49], the grower perceived water meter, as a practical, straightforward, and trustful sensor that could support efficient irrigation management. It also must be noted that in most cases the installation of water meter at the head of an irrigation system is a legal obligation.

For the evaluation of the DSS, its model was run using the actual irrigation applications that were performed during each irrigation period and then the derived by the model soil moisture time series were compared against those that were logged at the end of each day using the actual soil moisture sensors that were installed at the vineyard. During 2021, reasonable adjustments of the wetted area, irrigation efficiency, root depth and Kc values, were made to optimise the estimation of soil moisture by the DSS model against to those monitored by the sensors. Also, a secondary goal was for the model to recommend a number of irrigation events, close to those the grower had applied. That procedure was based on the resulting MAE and RMSE values of the DSS model timeseries.

During 2022, the optimised set of parameters of 2021 was used and only the phenological and managerial parameters (dates, periods, MAD and RF) were allowed to be changed. The grower did not have access to the DSS during each season but was briefed about the results at the end of each year.

Figure 2 presents the water balance parameters (effective rainfall, irrigation applications, soil moisture levels (Θs, FC, RAW and WP), the variation of the monitored soil moisture (soil moisture at the end of each day) and the variation of soil moisture as calculated by the DSS model (soil moisture at the end of each day) during the irrigation period

of 2021 for the plot irrigated using grower's experience (2021 GRO). The grower applied eight irrigations, while during the first two weeks of August, a continuous operation of the irrigation system occurred due to a damaged control valve. For the period between 20 April and 7 September, the total precipitation was 105 mm, while the grower applied 508 mm of water. The values of MAE and RMSE where 2.98% and 3.63%, respectively. On 7 September 2021, 18.5 t of grapes ha$^{-1}$ were harvested. The WPC for 2021 GRO plot, was calculated equal to 30.1 kg ha$^{-1}$/m$^3$ ha$^{-1}$.

**Table 2.** Parameters of the DSS for 2021 and 2022.

| Parameter | 2021 GRO | 2022 DSI | 2022 GRO |
|---|---|---|---|
| Potential effective rain coefficient | | 0.8 | |
| Total plot area (m$^2$) | 380 | 190 | 190 |
| Wetted area (m$^2$) | 220 | 110 | 110 |
| Irrigation efficiency | | 0.75 | |
| Maximum allowed depletion (MAD) | 0.45 | | 0.52 |
| Refill factor (RF) | 0.9 | | 0.5 |
| Estimated root depth (max) (m) | | 0.6 | |
| Kc off-season | | 0.1 | |
| Start of water balance season for each year | | 15/3 | |
| Planting date | 20/4 | | 15/4 |
| Kc on planting date | | 0.1 | |
| Kc stages duration (initial, development, mid-season, late-season) (days) | | 30 | |
| | | 60 | |
| | | 40 | |
| | 12 | | 32 |
| Kc (initial, mid-season, end) | | 0.4 | |
| | | 0.7 | |
| | | 0.4 | |
| Soil moisture at saturation (Θs, *v/v*) | | 0.45 | |
| Field capacity (FC, *v/v*) | | 0.34 | |
| Wilting point (WP, *v/v*) | | 0.13 | |

Figure 3 presents the water balance parameters (effective rainfall, irrigation applications, soil moisture levels (Θs, FC, RAW and WP), the variation of the monitored soil moisture (soil moisture at the end of each day) and the variation of soil moisture as calculated by the DSS's model (soil moisture at the end of each day) during the irrigation period of 2022 for the plot irrigated using the grower's experience (2022 GRO). The grower applied three (3) irrigations. For the period between 15 April and 24 September, the total precipitation was 131 mm, while the grower applied 219 mm of water. The values of MAE and RMSE where 3.22% and 4.06% respectively. On 24 September 2022, 12.6 t of grapes ha$^{-1}$ were harvested. The WPC for 2022 GRO plot, was calculated equal to 36.1 kg ha$^{-1}$/m$^3$ ha$^{-1}$.

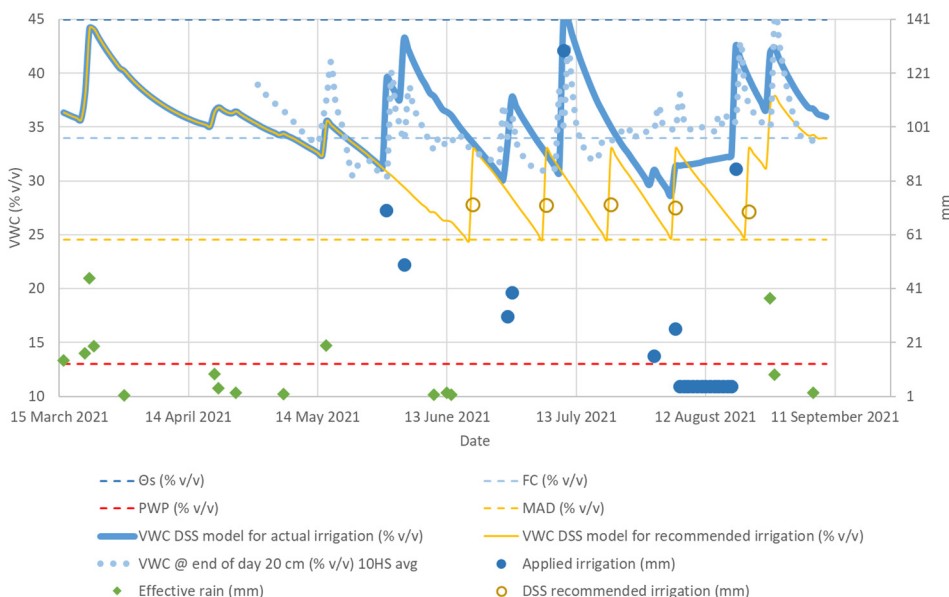

**Figure 2.** Monitored soil moisture and soil moisture generated by the DSS's model, based on actual irrigation applications, recommended irrigation applications and effective rain, during 2021 (2021 GRO plot). The symbol $v/v$ indicates that soil moisture is measured on volumetric basis.

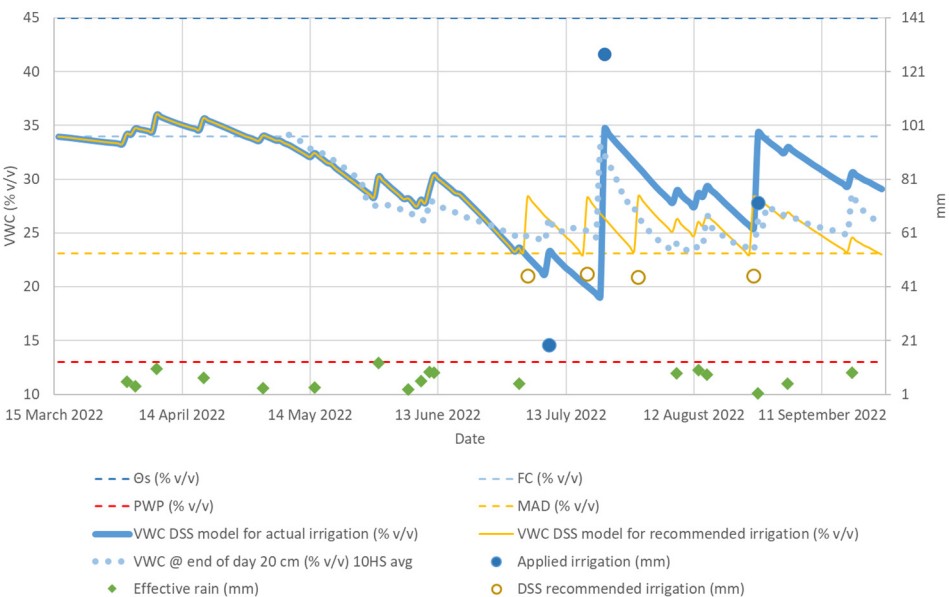

**Figure 3.** Monitored soil moisture and soil moisture generated by the DSS's model, based on actual irrigation applications, recommended irrigation applications and effective rain during 2022 (2022 GRO plot). The symbol $v/v$ indicates that soil moisture is measured on volumetric basis.

Figure 4 presents the water balance parameters (effective rainfall, irrigation recommendations, irrigation applications, soil moisture levels (Θs, FC, RAW and WP), the variation of the monitored soil moisture (soil moisture at the end of each day), of soil moisture as calculated by the DSS's model (soil moisture at the end of each day) taking into account the actual irrigations and the corresponding variation in that the DSS's recommendation would be strictly followed, during the irrigation period of 2022 for the plot irrigated using DSS's recommendations (2022 DSI). It must be noted that although the intention was to apply the recommendations of the DSS, the time windows of water availability, caused some deviations regarding the time of irrigation events. The DSS recommended four (4) irrigation events which would lead to the application of 182 mm of water. Following the DSS, four

(4) irrigations were applied (the first three of which could be considered as a set, as due to hydraulic problems of the irrigation system repairs were needed to complete the first irrigation). Furthermore, at the last irrigation, more water than what was recommended by the DSS was applied, due to a misunderstanding. For the period between 15 April and 24 September, the total precipitation was 131 mm, while the DSS team applied 272 mm of water. The values of MAE and RMSE where 3.05% and 3.70%, respectively. On 24 September 2022, 17.5 t of grapes ha$^{-1}$ were harvested. The WPC for 2022 DSI plot, was calculated equal to 43.5 kg ha$^{-1}$/m$^3$ ha$^{-1}$.

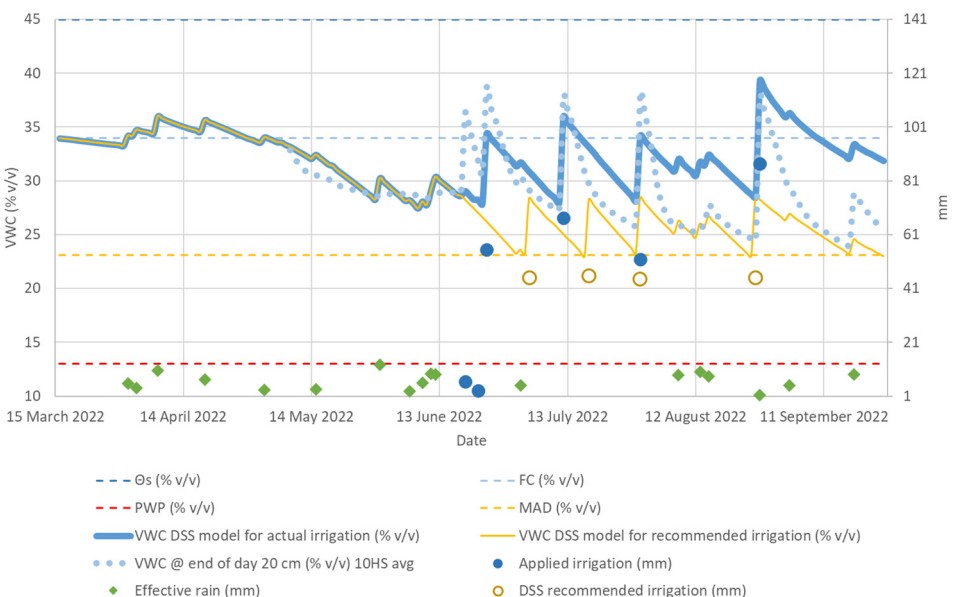

**Figure 4.** Monitored soil moisture and soil moisture generated by the DSS's model, based on actual irrigation applications, recommended irrigation applications and effective rain during 2022 (2022 DSI plot). The symbol $v/v$ indicates that soil moisture is measured on volumetric basis.

Measurements of LRWC (Figure 5), showed that during 2022, values for both DSI and GRO plots followed close routes, both at much higher of levels that would indicate water deficit [50,51].

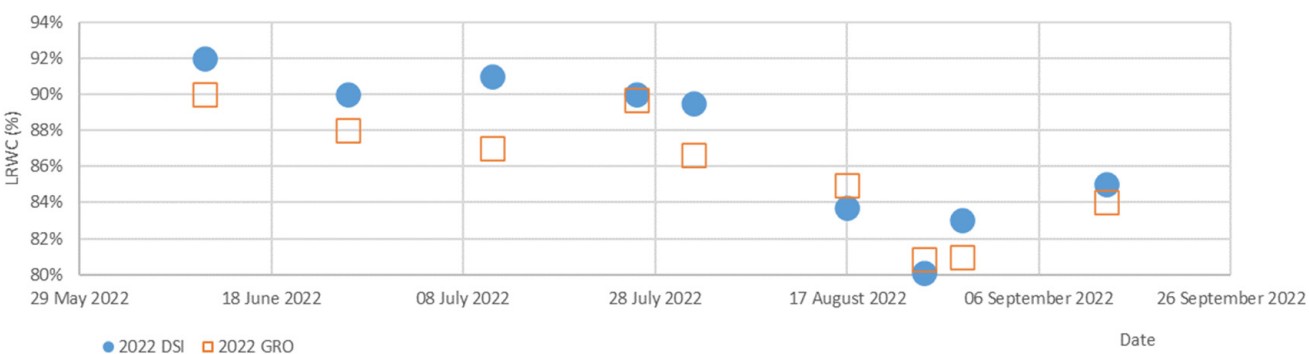

**Figure 5.** Course of midday Leaf Relative Water Content (LRWC) for 2022 GRO and 2022 DSI plots.

The fact that the grower was briefed about the results of the DSS and the potential impact to irrigation efficiency, probably affected the applied irrigation practice by the grower during the second year. The 2022 DSI plot received 24% more water than the 2022 GRO plot, but its larger yield, leaded to 20% raise of WPC. The results confirmed the relevant literature [1–9] regarding the achievement of high yield when irrigation is applied to grapevines and the application of the DSS resulted the reach of the highest

expected levels [19]. For the case of wine grapes, product quality in relation to water regime is typically of higher interest, and the scientific knowledge should be effectively integrated with the local experience in crop water management, to apply the best suited parameterisation of a decision support system, following the grower's perspectives [14]. For the present evaluation the main concern was the ability of the DSS to resemble readings of soil moisture sensors.

## 4. Conclusions

The main objectives of decision support systems for irrigation management are the reduction of irrigation water use and the relevant energy consumption along with the improvement of yield in both quantitative and qualitative terms. The DSS that was evaluated provides soil moisture estimations and irrigation recommendations based on the outcomes of a model that was based on the principles of UN FAO's paper 56, without the requirement for special monitoring hardware to be installed in each field.

The evaluation was made under commercial vineyard conditions and its main objective was the ability of the DSS to estimate soil moisture. A grower or an agronomic consultant can use generic parameters that are proposed by the DSS as a basis and alter only the parameters that could be estimated based on simple observations (wetted area, root depth, length of crop periods, irrigation system efficiency, MAD and RF). The results of the evaluation for the grape orchard of 'Vertzami' are very promising regarding the ability of the system's model to estimate the soil moisture at the field. The fact that the values of MAE and RMSE, which were used to evaluate the performance of the DSS to model soil moisture at the end of the day, ranged from 2.98–3.22% and 3.63% to 4.06%, respectively, during a two-year evaluation under various irrigation practices (GRO and DSI) and goals (MAD and RF), documents that the DSS could be efficiently used as an alternative to installation of soil moisture sensors at the field. The WPC for 2021 GRO plot, was calculated equal to 30.1 kg ha$^{-1}$/m$^3$ ha$^{-1}$, while for 2022 GRO plot, was calculated equal to 36.1 kg ha$^{-1}$/m$^3$ ha$^{-1}$. The WPC for 2022 DSI plot, was even better as it was calculated equal to 43.5 kg ha$^{-1}$/m$^3$ ha$^{-1}$ The expected—by previous research—potential of high yield when irrigation is applied to 'Vertzami' grapevines was confirmed and the highest reported levels were achieved by using the DSS. Future research will also include qualitative parameters of the yield.

The results showed that estimations of the DSS for rainfall for each area should be validated, because of the probable significant natural spatial variation of this agrometeorological variable, especially in small time base, could lead to misestimations. This is a probable pitfall for all relevant systems. In addition, the potential of large variations of readings from soil moisture probes in commercial open cropping systems was confirmed. Finally, the value of water meters as practical sensors which could support efficient irrigation management, is underlined. Application of such decision support systems that provide awareness of soil moisture levels for many fields in an area without the need for special hardware to be installed in each field could also support extensive trials of special irrigation management approaches such as regulated deficit irrigation (RDI) and partial root-zone drying (PRD). Furthermore, the application of such systems helps growers and agronomic consultants to document results, disseminate good practices and calculate environmental indices. In all cases, the operation of pilot evaluation fields is recommended for the development of proper sets of parameters for each crop and for technology demonstration purposes.

**Author Contributions:** I.L.T. and N.M. designed and supervised the research. I.L.T., N.M. and P.B. performed the research and analyzed the data. I.L.T. drafted the manuscript. N.M. and P.B. revised the manuscript. All authors have read and agreed to the published version of the manuscript.

**Funding:** The present work was co-financed by the European Union (European Regional Development Fund) and National Resources, under the operational program "Competitiveness, Entrepreneur-

ship and Innovation (EPAnEK)", "NSRF 2014–2020", Call 111: "Support for Regional Excellence (project: BIOFUSE/MIS 5047215).

**Data Availability Statement:** The data presented in this study are available on request from the corresponding author.

**Acknowledgments:** The authors would like to express their gratitude to Athanasios Tsolis, Grammenitsa, Arta, Greece for his cooperation and management of the experimental vineyard. Also, many thanks are owed to Aikaterini Glafki Apostolopoulou, Markos Giannelos, Eleni Lambraki, Dimitrios Giotis and Christos Koliopanos for their precious support at the vineyard.

**Conflicts of Interest:** The authors declare that they have no known competing financial interest or personal relationships that could have appeared to influence the work reported in this paper.

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
