# Peer review of "Application of a Generic Participatory Decision Support System for Irrigation Management for the Case of a Wine Grapevine at Epirus, Northwest Greece"

_horticulturae, doi:10.3390/horticulturae9020267_

Round 1

Reviewer 1 Report

Please find the comments in the attached file. 

Author Response

Thank you very much for your useful comments. Bellow you will find our replies.

Point 1: The introduction can be improved. 

Response 1:  The introduction was enriched, and more information is provided regarding the framework and the research objectives of the study, which are the reduction of irrigation water use and the relevant energy consumption along with the increase of the yield by applying a DSS that does not require special monitoring hardware to be installed in each field.

Point 2: Line 54-56 The aim of the article is to ‘conclude whether the DSS could be used instead of soil moisture sensors providing an adequate solution for growers and if its application could improve water productivity’. I suggest a description of the advantages and disadvantages of ‘soil moisture sensors’ or the current status of their use, and a description of the reasons of using DSS.

Response 2: More information about the advantages and disadvantages of ‘soil moisture sensors’ and the status of their use, along with extra references has been added at section 3.2.

Point 3:  The description of the methods can be improved. / I propose to give a brief description of the background, concept and main functions of ‘decision support system (DSS)’. Or you can draw simple flowcharts, such as 'DSS basic structure diagram', 'DSS system structure diagram', etc. to describe its function or working principle.

Response 3:  The “Materials and Methods” section was reformed regarding the description of the DSS and new information was added. Documentation and analytical flowcharts of the structure and the functions of the DSS is provided by Malamos, et al. 2016. A relevant sentence has been added at the Materials and Methods section, along with the reference. Also, information regarding measurements of midday Leaf Relative Water Content (LRWC) was added.

Point 4: The Results section can be improved.

Response 4:  The sections for agrometeorological parameters, soil moisture monitoring and DSS performance were updated. New information and relevant references were added and discussed in the framework of the results obtained from the presented evaluation.

Point 5: Conclusions: I suggest improvements to the description of the conclusion section. Please try to focus on the principal findings. In response to the ‘aim of the evaluation’ mentioned earlier, I suggest a clear and concise account of the findings obtained. In addition, is it possible to describe the prospective applications of DSS and the limitations that exist.

Response 5:  The Conclusions section was extensively updated. Information regarding all the findings, limitations and prospective applications was added.

Reviewer 2 Report

What recommendations do the authors have to reduce the difference between the actual and estimated productivity of vineyards?

It is necessary to measure the transpiration of grape bushes depending on watering, as one of the most important indicators.

In addition to the depth of penetration of grape roots into the soil, it is also necessary to measure the volume of the root system depending on irrigation.

It is advisable to measure the processes of photosynthesis and respiration of the grape bush during irrigation, since these indicators are important for plant productivity.

It is necessary to provide a list of references in accordance with the rules of the journal.

Author Response

Thank you very much for your useful comments. Bellow you will find our replies.

Point 1: The introduction can be improved. 

Response 1:  The introduction was enriched, and more information is provided regarding the framework and the research objectives of the study, which are the reduction of irrigation water use and the relevant energy consumption along with the improvement of yield by applying a DSS that does not require special monitoring hardware to be installed in each field.

Point 2: The description of methods can be improved. 

Response 2:  The “Materials and Methods” section was reformed regarding the description of the DSS and new information was added. Also, information regarding measurements of midday Leaf Relative Water Content (LRWC) was added.

Point 3: What recommendations do the authors have to reduce the difference between the actual and estimated productivity of vineyards?

Response 3:  The results confirmed the relevant literature regarding the achievement of high yield when irrigation is applied to ‘Vertzami’ grapevines and the application of the DSS resulted the reach of the highest expected levels. This information along with the relevant reference was added to the manuscript.

Point 4: It is necessary to measure the transpiration of grape bushes depending on watering, as one of the most important indicators.

Response 4:  The only relevant available data that are available, are measurements of midday Leaf Relative Water Content (LRWC) for each treatment during 2022. This information has been incorporated in the manuscript and presented in Figure 5.

Point 5: In addition to the depth of penetration of grape roots into the soil, it is also necessary to measure the volume of the root system depending on irrigation.

Response 5:  The depth of the root system was not measured. The estimation of the volume of soil that was wetted by the irrigation system was based on measurements of the wetted soil surface area that was achieved by the operation of the drippers and the grower’s goal of soil depth for which the moisture should be controlled. Values for both parameters are provided in Table 2 of the manuscript.

Point 6: It is advisable to measure the processes of photosynthesis and respiration of the grape bush during irrigation, since these indicators are important for plant productivity.

Response 6: No relevant measurements were made.

Point 7: The Results section can be improved.

Response 7:  The sections for agrometeorological parameters, soil moisture monitoring and DSS performance were updated. New information and relevant references were added and discussed in the framework of the results obtained from the presented evaluation.

Point 8: It is necessary to provide a list of references in accordance with the rules of the journal.

Response 8:  The list of references was corrected following the formatting guidelines of the journal.

Reviewer 3 Report

Comments and Suggestions for Authors

The paper titled ‘Application of a generic participatory decision support system  for irrigation management for the case of a wine grapevine at  Epirus, northwest Greece’ presents results obtained from a 2-year field experiment, which aimed to improve irrigation management methods based on decision support system for irrigation management (DSS).  The obtained results are representative for the local conditions.

The abstract is informative, however there is no clear research objective - what for the research was conducted? to safe the water, energy, to avoid salinization, to increase the yield

The manuscript is clearly and concisely written. The method are well described and results are presented in detail.

The range of references is appropriate and properly cited.

However there is a week point in the manuscript, which is no results discussion. Authors do not refer their results to any other scientific studies, which in any scientific paper, this academic discussion has to be done. I suggest the Authors to complete this part of the manuscript.

In the conclusion section I suppose to find such information, how this moisture monitoring will save water discharge, ore energy, or if it will protect against salinization or erosion of soil, but I did not.

I also found some minor incorrections, which needs to be changed/corrected

L. 32, 34, 36: a number of references which are subsequent (more than 2 items), should be written with a hyphen e.g. [1–4], [5–9], [10–12] etc.

L. 64: there is ‘tn  ha-1’ if you mean tons per hectare it should be changed into ‘t·ha-1’?

L. 71 , 72:  the unit (% kg kg-1) ?  are you sure it is justified?

L. 74: (% m m-3)? I think there is too much units in the parenthesis (see the above comment)

L. 150  and 151: there is ‘mm or rain’ – change into ‘mm of rain’

L. 156 and 157: in the Figure 1 appear characters as (a) and (b), which are not explain below the graphical objects in the Figure 1 description

L. 177:  please check the units in the Table 1 (see the comment L.74)

L. 227: there is ‘tn’ - should be ‘t’  (see comment  L. 64)

Also I would like to be explained, if DDS-based irrigation was fully automated ( I mean automatic opening and closing of the water valve based on DDS recommendations) or did it require human intervention?

Author Response

Thank you very much for your useful comments. Bellow you will find our replies.

Point 1: The introduction can be improved.

Response 1:  The introduction was enriched, and more information is provided regarding the framework and the research objectives of the study, which are the reduction of irrigation water use and the relevant energy consumption along with the increase of the yield by applying a DSS that does not require special monitoring hardware to be installed in each field.

Point 2: The Results section must be improved. / However there is a week point in the manuscript, which is no results discussion. Authors do not refer their results to any other scientific studies, which in any scientific paper, this academic discussion has to be done. I suggest the Authors to complete this part of the manuscript.

Response 2:  The sections for agrometeorological parameters, soil moisture monitoring and DSS performance were updated. New information and relevant references were added and discussed in the framework of the results obtained from the presented evaluation.

Point 3: The support of conclusions by the results must be improved. / In the conclusion section I suppose to find such information, how this moisture monitoring will save water discharge, or energy, or if it will protect against salinization or erosion of soil, but I did not.

Response 3:  The Conclusions section was extensively updated. Information regarding all the findings, limitations and prospective applications was added.

Point 4: L. 32, 34, 36: a number of references which are subsequent (more than 2items), should be written with a hyphen e.g. [1–4], [5–9], [10–12] etc.

Response 4: All necessary corrections were made throughout the manuscript.

Point 5: L. 64: there is ‘tn΄ ha-1’ if you mean tons per hectare it should be changed into‘t·ha-1’? and L. 227: there is ‘tn’ - should be ‘t’ (see comment L. 64)

Response 5: Indeed, it is tons per hectare, and tn was corrected throughout the manuscript.

Point 6: L. 71 , 72: the unit (% kg kg-1)? are you sure it is justified?

Response 6: It is not wrong, but it is not the typical presentation of gravitational proportions of the contents of a soil sample. I deleted kg kg-1.

Point 7: L. 74: (% m m-3)? I think there is too much units in the parenthesis (see the above comment) and L. 177: please check the units in the Table 1 (see the comment L.74)

Response 7: In both cases, the m3 m-3 part of the unit was deleted and the indication volumetric was used to define the basis of percentage value.

Point 8: L. 150 and 151: there is ‘mm or rain’ – change into ‘mm of rain’

Response 8: Corrected.

Point 9: L. 156 and 157: in the Figure 1 appear characters as (a) and (b), which are not explain below the graphical objects in the Figure 1 description

Response 9: Indications for (a) and (b) were added to the legend of Figure 1.

Point 10: Also I would like to be explained, if DDS-based irrigation was fully automated (I mean automatic opening and closing of the water valve based on DDS recommendations) or did it require human intervention?

Response 10: The recommendations have to applied manually as the DSS does not support automatic control of water valves. A relevant clarification has been added at “Materials and Methods”.

Round 2

Reviewer 3 Report

Thank you the Authors for introducing all the essential changes to the manuscript. I believe it improved its quality.